

# Detecting the outbreak of influenza based on the shortest path of dynamic city network

Yingqi Chen[1], Kun Yang[1], Jialiu Xie[2], Rong Xie[3], Zhengrong Liu[4], Rui Liu[4] and Pei Chen[4]

[1] School of Computer Science and Engineering, South China University of Technology, Guangzhou, Guangdong, China
[2] Department of Biostatistics, University of North Carolina at Chapel Hill, Chapel Hill, NC, United States of America
[3] School of Information, Guangdong University of Finance and Economics, Guangzhou, Guangdong, China
[4] School of Mathematics, South China University of Technology, Guangzhou, Guangdong, China

## ABSTRACT

The influenza pandemic causes a large number of hospitalizations and even deaths. There is an urgent need for an efficient and effective method for detecting the outbreak of influenza so that timely, appropriate interventions can be made to prevent or at least prepare for catastrophic epidemics. In this study, we proposed a computational method, the shortest-path-based dynamical network marker (SP-DNM), to detect the pre-outbreak state of influenza epidemics by monitoring the dynamical change of the shortest path in a city network. Specifically, by mapping the real-time information to a properly constructed city network, our method detects the early-warning signal prior to the influenza outbreak in both Tokyo and Hokkaido for consecutive 9 years, which demonstrate the effectiveness and robustness of the proposed method.

Corresponding authors
Rui Liu, scliurui@scut.edu.cn
Pei Chen, chenpei@scut.edu.cn

## INTRODUCTION

Vast amounts of time and resources are being invested in planning for the next influenza pandemic. However, despite the great efforts of prevention and control, seasonal influenza remains a significant cause of morbidity and mortality worldwide, particularly among persons aged ≥65 years and <2 years and those with medical conditions that confer high risk for complications from influenza (*Charu et al., 2011*; *CDC, 2013*). Specifically, the annual averages of 226,054 (range, 54,523–430,960) primary and 294,128 (range, 86,494–544,909) any listed respiratory and circulatory hospitalizations were associated with influenza virus infections (*Thompson et al., 2004*). Furthermore, the total economic burden of annual influenza epidemics amounted to $87.1 billion in US (*Molinari et al., 2007*).

As the influenza pandemic is a great threat to both public health and social economics, many studies were devoted to influenza control strategy, including the social distancing interventions to restrain influenza spread (*Kelso, Milne & Kelly, 2009*), the collaboration

among fields involving etiology, epidemiology, clinical practices, preventive medicine, and molecular biological engineering (*Zhong & Zeng, 2003*), and mathematical models such as a large-scale stochastic simulation model to investigate the spread of a pandemic strain of influenza virus (*Germann et al., 2006*), and logistic regression models of influenza (*Pfeiffer et al., 2007*; *Boivin et al., 2000*). There is no doubt that these theoretical and practical efforts benefit influenza control. However, it is difficult to predict the outbreak of influenza due to the complexity of its temporal and spatial characteristics in the evolution and transmission processes. Therefore, to accurately signal the influenza outbreak, it should take regional geographic information, transportation, the size of population and the number of clinics, real-time clinic visiting data and other information into consideration.

The outline of our study was shown in Fig. 1. The aim of this study was to solve the problem that predicting the outbreak of influenza is difficult, so as to predict influenza outbreak accurately, and save economic losses and human lives (Fig. 1C). In this study, we present a novel computational method, the shortest-path-based dynamical network marker (SP-DNM), to detect the early-warning signal of influenza outbreak. Specifically, the dynamical progression of the influenza outbreak is divided into three states, that is, the normal state, the pre-outbreak state and the outbreak state (Fig. 1B) (*Liu et al., 2012*; *Scheffer et al., 2009*; *Venegas et al., 2005*). Based on SP-DNM, a city network was constructed by combining the geographically adjacent information, transportation, population mobility and the number of clinics of each city district. It was found that the dynamical change of the city network reflects the severity of influenza. More importantly, an index that quantitatively measures the shortest paths of the city network provides an accurate way to detect the early-warning signal to the outbreak of influenza. This method has solid theoretical background, that is, the dynamic network marker (DNM), which theoretically proves that there is a dominant group of variables (DNM) satisfying three generic properties when a system is in a critical state, that is, (1) the correlation between any pair of members in the DNM group rapidly increases; (2) the correlation between one member of the DNM group and any other non-DNM member rapidly decreases; (3) the standard deviation or coefficient of variation for any member in the DNM group drastically increases (*Liu et al., 2014*; *Liu, Aihara & Chen, 2013a*). The DNM method has been applied in cell differentiation (*Chen et al., 2015*; *Richard et al., 2016*), cancer (*Lesterhuis et al., 2017*; *Liu et al., 2019*) and diabetes (*Liu et al., 2013*; *Li et al., 2014*), achieving satisfactory result in many field. Based on above three generic conditions, SP-DNM is capable of quantitatively measuring the criticality of the influenza outbreak, that is, the dynamically significant changes of the shortest paths of the city network. Specifically, in Fig. 1B, the SP-DNM method can easily distinguish pre-outbreak states by exploring the critical spatial information from longitudinal high-dimensional historical records so that timely proactive action such as vaccine, distance isolation can be performed to prevent or at least mitigate the influenza outbreak (*Kelso, Milne & Kelly, 2009*; *Zhong & Zeng, 2003*). It was found that the sudden change of shortest path in a city network can accurately identify a pre-outbreak state, or equivalently, the end of a normal state of the network system, thus detect the early-warning signal of the upcoming critical transition into a serious and irreversible outbreak state. In this study, a weight was assigned to each edge of the city network,

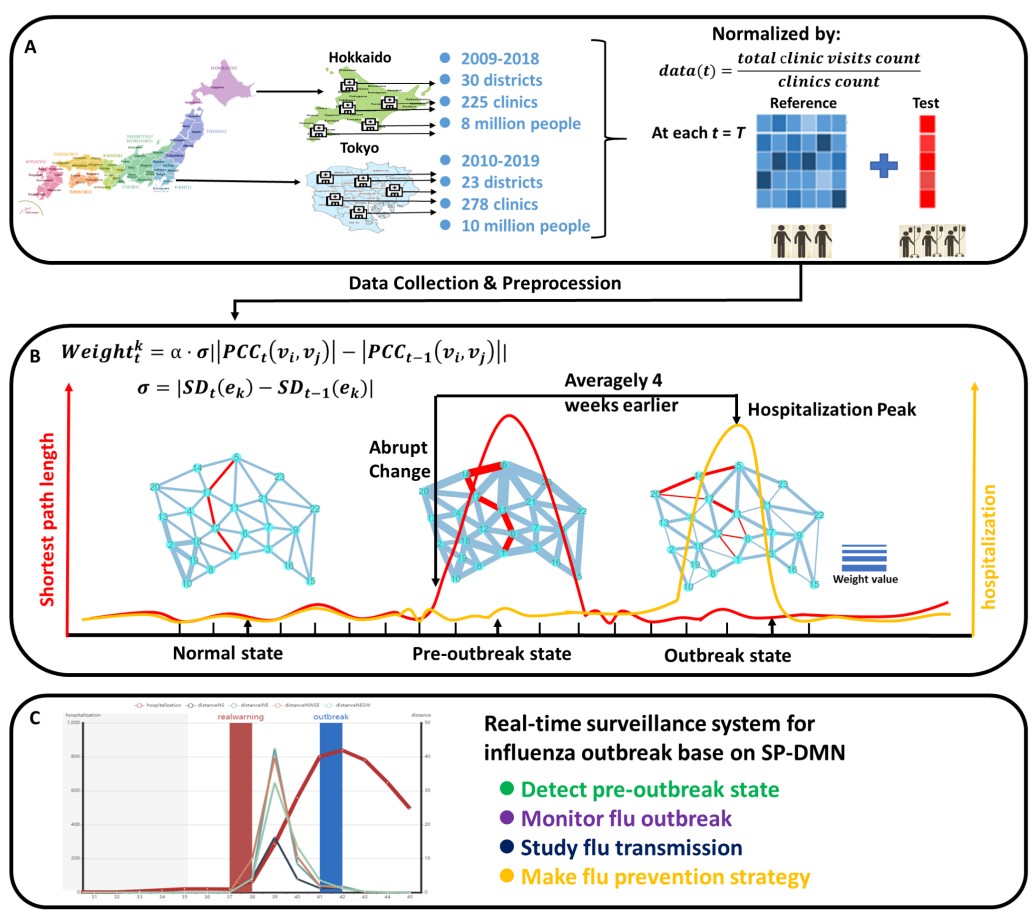

**Figure 1 The outline of the proposed SP-DNM method.** (A) In this study, the clinic visiting records caused by influenza in Tokyo and Hokkaido were collected, covering 23 districts in Tokyo, 30 cities in Hokkaido and 503 clinics. The data was normalized by the count of clinics in its region. (B) A city network was constructed based on the geographically adjacent information of the city districts. In the city network, weight of edges can be measured by Pearson correlation coefficient (PCC) and standard deviation (SD) which indicate pre-outbreak state according to data we collected. Then, the Dijkstra algorithm is implied into this weighted network to acquire the SP-DNM scores, whose rapid increase means that this network is closed its pre-outbreak state. (C) Conclusion: By using SP-DNM, we can establish a real-time system to monitor influenza outbreak, detect pre-outbreak state, study influenza transmission and make flu prevention strategy to save people's lives.

which was the correlation between the numbers of clinic visits of two adjacent districts. Then, the four shortest paths were calculated and monitored in the dynamic city network, i.e., (1) northernmost to southernmost; (2) westernmost to easternmost; (3) northwest to southeast; (4) northeast to southwest. A composite index based on the dynamical changes of these four shortest paths was employed to quantitatively measure the criticality of the city network. The abrupt increase of the composite index provides the early-warning signal of an impending critical transition into the influenza outbreak.

To demonstrate the effectiveness of our method, we applied SP-DNM to the historical records in Tokyo and Hokkaido for consecutive 9 years, from 2010 to 2019 in Tokyo and

from 2009 to 2018 in Hokkaido. SP-DNM accurately predicted each influenza outbreak averagely 4 weeks ahead in Tokyo and 6 weeks ahead in Hokkaido. Moreover, by surveilling the dynamic changes of the city network, it provides a new approach to study the epidemic spread in a city.

## MATERIALS & METHODS

SP-DNM has solid theoretical background, i.e., the DNM theory (*Liu et al., 2015*; *Liu, Chen & Chen, 2020*), according to which it divides the progression of epidemic dynamics into three state, (1) a normal state; (2) a pre-outbreak state; (3) an outbreak state (*Chen et al., 2012*) and provides some statistical properties to identify the pre-outbreak state which is between normal state and outbreak state. These statistical properties are listed below (*Chen et al., 2012*) and data below, that is $x_1, x_2, y_1, y_2$, are high-dimensional with longitudinal historical record.

1. $SD(x_1)$ increases sharply, where $x_1$ is historical data of DNM member in network, SD represents the standard deviation of variables $x_1$.
2. $PCC(x_1, x_2)$ increases sharply, where $x_1$ and $x_2$ are historical data of DNM member in network, PCC represents the Pearson correlation coefficient between two variables.
3. $PCC(x_1, y_1)$ decreases sharply, where $x_1$ is historical data of DNM member in network while $y_1$ represents historical data of non-DNM member in network.
4. $SD(y_1)$ and $PCC(y_1, y_2)$ would not have any remarkable change or be in specific rule, where $y_1$ and $y_2$ represent historical data of non-DNM member in network.

By using these statistical properties above, it is possible to differentiate normal state and pre-outbreak state, which means that the pre-outbreak state, which is unstable and sensitive to perturbations, could be detected. Once the pre-outbreak state is detected, proactive strategies can be exploited to reverse it to the normal state (*Mather, 1982*).

In traditional DNM method, researchers use $I_{DNM}$ score to detect pre-outbreak state by combining statistical properties (1), (2) and (3):

$$I_{DNM} = \frac{PCC_{in}}{PCC_{out}} SD_{in}$$

where $PCC_{in}$ represents correlation between two DNM member, $PCC_{out}$ represents correlation between a DNM member and a non-DNM member, $SD_{in}$ represents standard deviation of a DNM member. It is noted that such DNM method is with solid theoretical background and thus has a series of following modified approaches and applications to real-world cases. For example, an algorithm called Landscape DNM method was developed which employs local-landscape score on the basis of the DNM statistical properties (*Chen et al., 2019*). The Markov model was applied to describe the three states of disease progression, which enhance the accuracy of obtaining information from statistical properties (*Chen et al., 2016*; *Chen et al., 2017*), etc.

### Algorithm

The overview of our algorithm was presented in Fig. 2 and code was shown in Algorithm 1. In a city network with longitudinal clinic-visiting record, the weight of edge can be

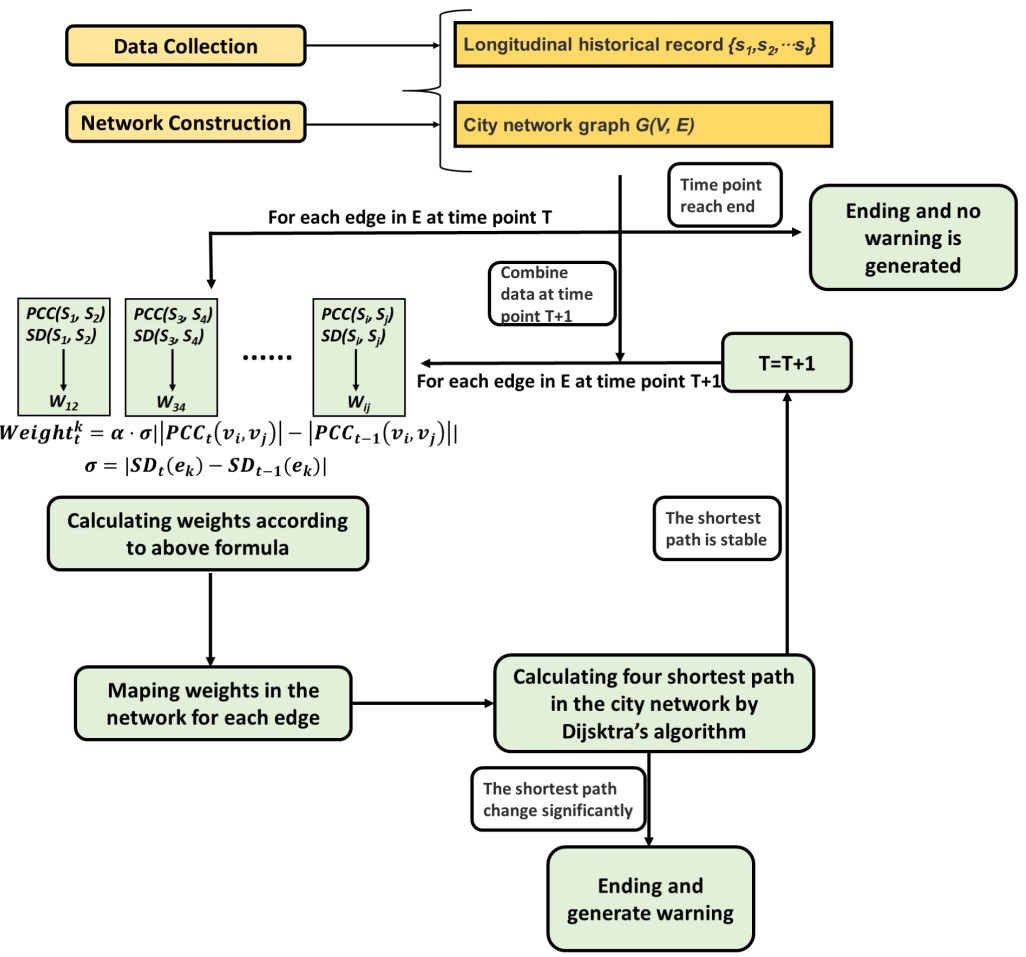

**Figure 2 The diagram interpretation of SP-DNM.** Including three steps, (1) constructing network by using gene information or geographical information, (2) constructing the weighted network by calculating Pearson correlation coefficient and standard deviation of each edge, (3) detecting the pre-outbreak state by surveilling sudden change of the shortest path in the network.

measured by the Pearson correlation coefficient and standard deviation of its associated nodes. Based on the DNM theory, the transformation of different state of flu outbreak progression can be described by a city network with changing weights. In other words, the sharp increase of weight indicated that the pre-outbreak of influenza is coming. In order to avoid the abnormal signals caused by sudden local surges of weights, the Dijkstra algorithm was employed in this city network to calculate the shortest path. The procedure of the SP-DNM method was described as follows.

---

**Algorithm 1** SP-DNM

---

**Input:** $S\{s_1,s_2,...,s_t\}$: Longitudinal historical hospitalization record caused by flu;

$G(V,E)$: City network graph modeled according to geographical information

**Output:** $T$: Real early-warning signal of flu

$T_m$: Mild early-warning signal of flu

  1:  initialization: set $T = 0$, $T_m = \emptyset$; set $d_0(i) = d_{-1}(i) = \infty, i \in \{1,2,3,4\}$ ;

       set $PCC_0(i,j) = SD_0(k) = 0$, for each $e_{ij} \in E$, $v_k \in V$

  2:  **while** $d_T(i) < d_{T-1}(i), i \in \{1,2,3,4\}$ **do**

  3:      $T = T + 1$

  4:      **for** edge $e_{ij} \in E$ **do**

  5:          $\delta = ||SD_t(v_i,v_j)| - |SD_{t-1}(v_i,v_j)||$ where $SD_t(v_i,v_j) = |SD_t(i) + SD_t(j)|/2$

  6:          Weight the edge $e_{ij}$ with $W_{ij}^t = \delta ||PCC_t(v_i,v_j)| - |PCC_{t-1}(v_i,v_j)||$

  7:      **end for**

  8:      Calculate the four shortest paths' weight sum $d_T(i) = Dijkstra(G(V,E))$

  9:      **if** three shortest paths satisfy the condition $d_T(i) \geq d_{T-1}(i)$ **then**

10:         $T$ add to the set $T_m$

11:      **end if**

12:  **end while**

13:  return $T$, $T_m$

---

### Constructing city network

The dynamic city network is the premise of our method. For the application of DNM method in gene expression, gene network was constructed by mapping genes to protein-protein interaction network and extracting the central gene's associated genes as its first-order neighbors. For detecting influenza outbreak, it is reasonable to use geographical and transportation information to construct a global city network. The city network was formed by geographical and transportation information. In the network, each node represents a district, while each edge represents the main traffic route connecting two districts. As shown in Fig. 3A, there are 23 different regions in Tokyo. Correspondingly, there are 23 nodes and 53 edges in the Tokyo city network according to its real geographical position and traffic route.

In addition, the dynamical change of transportation among the cities has been taken into consideration. According to the passenger flow of the main traffic route or metro station, a coefficient parameter $\alpha$ has been assigned to corresponding edge, that is, such coefficient for the maximum passenger flow is set as 1, and the other coefficient is the percentage of the corresponding passenger flow in the maximum. Due to the yearly dynamic change of passenger flow, the city network alters year by year.

### Constructing the weighted city network

A city network can be described as a graph $G = (V,E)$, where $V = \{v_1,v_2,\ldots,v_n\}$ is the set of vertexes of city network and $E = \{e_1,e_2,\ldots,e_m\}$ is the set of edges of city network. There are the following procedures implied into the city network.

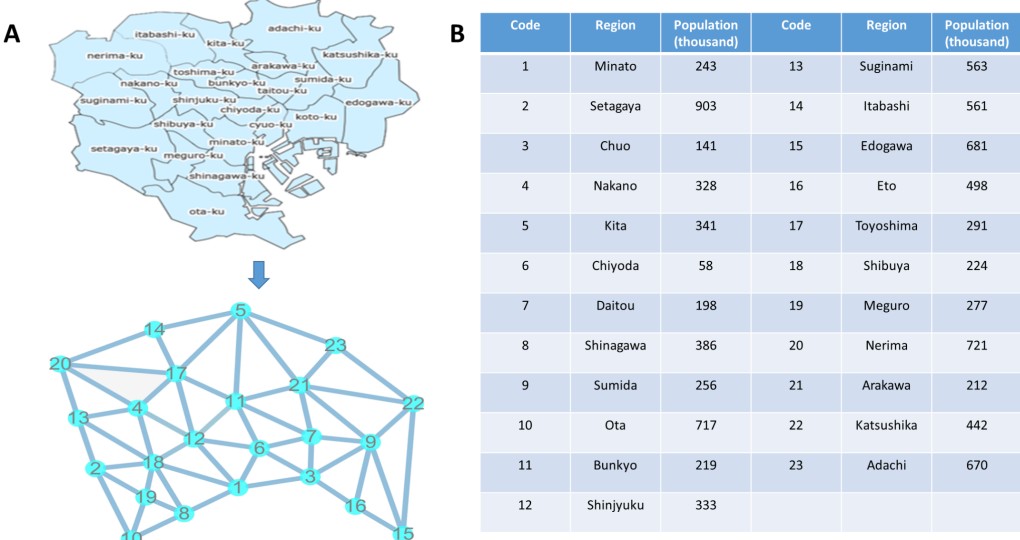

**Figure 3  Map and network of Tokyo.** (A) Original map of Tokyo and its constructed network. (B) The name of 23 regions and its corresponding code and population in the table.

First, based on the longitudinal historical records, we considered the data per week as a sample. In other words, for each vertex $v_i$ at time point $t$, there are a series of time series data $\{s_1, s_2, \ldots, s_t\}$.

Second, for each edge $e_k$ of the city network at week $t$, assign it a weight $Weight_t^k$ with the correlations of its two vertexes $v_i, v_j$:

$$Weight_t^k = \alpha \cdot \sigma \left| \left| PCC_t \left( v_i, v_j \right) \right| - \left| PCC_{t-1} \left( v_i, v_j \right) \right| \right|$$

where $PCC_t \left( v_i, v_j \right)$ represents the Pearson correlation coefficient of the two vertexes $v_i, v_j$ at week $t$, $PCC_{t-1} \left( v_i, v_j \right)$ represents the Pearson correlation coefficient of the two vertexes $v_i, v_j$ at week $t-1$, the year-varying parameter $\alpha$ represents a coefficient corresponding to the yearly transportation volume between two districts and the parameter $\sigma$ is expressed by the following equation:

$$\sigma = \left| \frac{SD_t \left( v_i \right) + SD_t \left( v_j \right)}{2} - \frac{SD_{t-1} \left( v_i \right) + SD_{t-1} \left( v_j \right)}{2} \right|$$

where $SD_t \left( v_i \right)$ represents the standard deviation of the data of vertex $v_i$ at week $t$, and $SD_{t-1} \left( v_i \right)$ represents the standard deviation of the data of $v_i$ at the previous week $t-1$.

### Detecting the pre-outbreak state

In order to accurately and quantitatively reveal the dynamical changes caused by the longitudinal historical records, it is required to obtain the shortest paths of the city network, which is sensitive to the local abnormal variation of the city network. In this study, the Dijkstra algorithm was implied to obtain the city network's shortest paths to detect the pre-outbreak of influenza outbreak.

Then, to reflect the global change of the network through the shortest paths, four shortest paths were defined, this is, the northernmost node to the southernmost node, the

westernmost node to the easternmost node, the northwest node to the southeast node, the northeast node to the southwest node. The length of each shortest path was regard as the SP-DNM scores:

$$L_t^i = \sum_{j=1}^{K^i} Weight_j^i, \quad i \in \{1,2,3,4\}$$

where $L_t^i$, $K^i$ and $Weight_j^i$ represents the length, the edge numbers and the weight of each edge of $ith$ shortest path respectively at the week $t$.

According to the DNM theory, the SP-DNM scores, which was based on the standard deviations of these vertexes in the city network and their Pearson correlation coefficients, could be employed to quantitatively describe changes in the city networks caused by influenza, thereby to detect the pre-outbreak state.

In this study, a 3-fold change threshold was implied to detect the signal of tipping point. That is, we calculate the SP-DNM scores for the city network at a week $t$. If the SP-DNM scores satisfied the following condition, the time point $t$ was regarded as the tipping point:

$$L_t^i \geq 3*L_{t-1}^i, \ i \in \{1, 2, 3, 4\}.$$

Otherwise, week $t$ is considered as in the normal state, and the data derived from week $t$ are also comprised in the control samples. Then, the next timepoint $t+1$ is selected as the candidate to carry on our algorithm until the pre-outbreak was detected.

## RESULTS

In this section, we applied the SP-DNM method to the historical records in Tokyo and Hokkaido for consecutive 9 years, from 2010 to 2019 in Tokyo and from 2009 to 2018 in Hokkaido. It is showed that SP-DNM accurately predicted each influenza outbreak averagely 4 weeks ahead in Tokyo and 6 weeks ahead in Hokkaido.

The data of Tokyo was obtained from Tokyo Metropolitan Infectious Disease Surveillance Center (http://survey.tokyo-eiken.go.jp/epidinfo/weeklyhc.do), including the number of clinic visits every week in a year from 23 region. The data of Hokkaido was obtained from Hokkaido Infectious Disease Surveillance Center (http://www.iph.pref.hokkaido.jp/kansen/501/data.html), including the number of clinic visits every week in a year from 30 region. During 2010–2019 (in Tokyo) and 2009–2018 (in Hokkaido), the number of clinic visits per week in each region has been collected. These dataset span 53 districts and nine years, involving 20 million people. Both the two datasets have been normalized by dividing the number of clinics in a region. The transportation data of Tokyo was obtained from Tokyo statistical yearbook (https://www.toukei.metro.tokyo.lg.jp/tnenkan/tn-index.htm) and the transportation data of Hokkaido was obtained from Hokkaido District Transport Bureau (https://wwwtb.mlit.go.jp/hokkaido/kakusyu/toukei/index.html).

### Detecting the pre-outbreak state in Tokyo
According to the proposed SP-DNM method, the following procedures were carried out to identify the pre-outbreak state of flu outbreak in Tokyo. Firstly, the city network shown in Fig. 3A was constructed according to the geographical location and transportation. Additionally, the regions and their corresponding codes and population were listed in Fig. 3B. Secondly, weights were assigned to each edge as showed in the section of Materials

and Methods. It should be noted that, because influenza outbreak often occurs between the end of one year and the beginning of the next year, we regard a year in our algorithm is from April 1 to March 31 of the following year. Thirdly, Dijkstra algorithm was applied to the city network to obtain its four shortest paths and figure out the SP-DNM scores. Four shortest paths we defined are: (1) node 5, *Kita-ku* to node 1, *Minato-ku*, represents the northernmost node to the southernmost node; (2) node 13, *Suginami-ku* to node 22, *Katsushika-ku*, represents the westernmost node to the easternmost node; (3) node 20, *Nerima-ku* to node 15, *Edogawa-ku*, represents the most northwest node to the most southeast node; (4) node 23, *Adachi-ku*, to node 10, *Setagaya-ku*, represents the most northeast node to the most southwest node. Finally, the SP-DNM scores were employed to determine whether the city network has reached its pre-outbreak state.

The prediction results of influenza outbreak in Tokyo has been shown in Fig. 4. The early-warning signals of flu were detected by our method before influenza outbreak from 2010 to 2019. It is clear that the overall dynamical trend of SP-DNM curve was ahead of the clinic-visiting curve, which indicates that the early-warning role of the SP-DNM method. The red bars represent real warnings, which are indicated if all SP-DNM scores passed the 3-fold change threshold. It is seen that these real warnings were ahead of blue bars which represent the influenza outbreak, demonstrating the accuracy of our method. After red bar, the pre-outbreak state, the number of clinic visits increase rapidly and reach the peak after 3–6 weeks. The results presented in the figure reveal that our method identifies the early warning of influenza outbreak accurately with an average of 4-week window ahead, which provides adequate time to control influenza outbreak. Besides, yellow bars represent mild warnings, which are indicated if a proportion of 3/4 SP-DNM scores (three shortest paths but not all) satisfied the 3-fold change threshold. It is seen that after the mild warnings the clinic-visiting count increases slowly in several weeks.

Based on the severity of flu spread, we classify our warning signal into two classes, the real warning and mild warning. It is seen in Fig. 4 that from 2016 to 2019after the real warning time points, the clinic-visiting count increases sharply, which peaked in a few weeks. After the mild warning time points, the clinic-visiting count increases slowly in several weeks. Therefore, the mild warning signal means that influenza only spreads in some marginal region and most core regions are still in control for the flu infection. In other words, it is beneficial to trace the city network to find the region where influenza strikes and make proactive strategy.

As the results illustrated, SP-DNM predicts the pre-outbreak state accurately. It is worth noting that it is meaningful to remain an average of 4-week window without drastic fluctuation to make proactive strategy. When the prediction window is more than six weeks, the situation of flu outbreak may change over time, resulting in that the flu prevention and control strategies need to be constantly changed causing a lot of human and financial loss. In the meanwhile, when the prediction window is less than two weeks, it is extremely tough to make effective flu precautions in a limited time. In general, the prediction window of four week in average is relatively suitable to make the timely and efficacious influenza control measures.

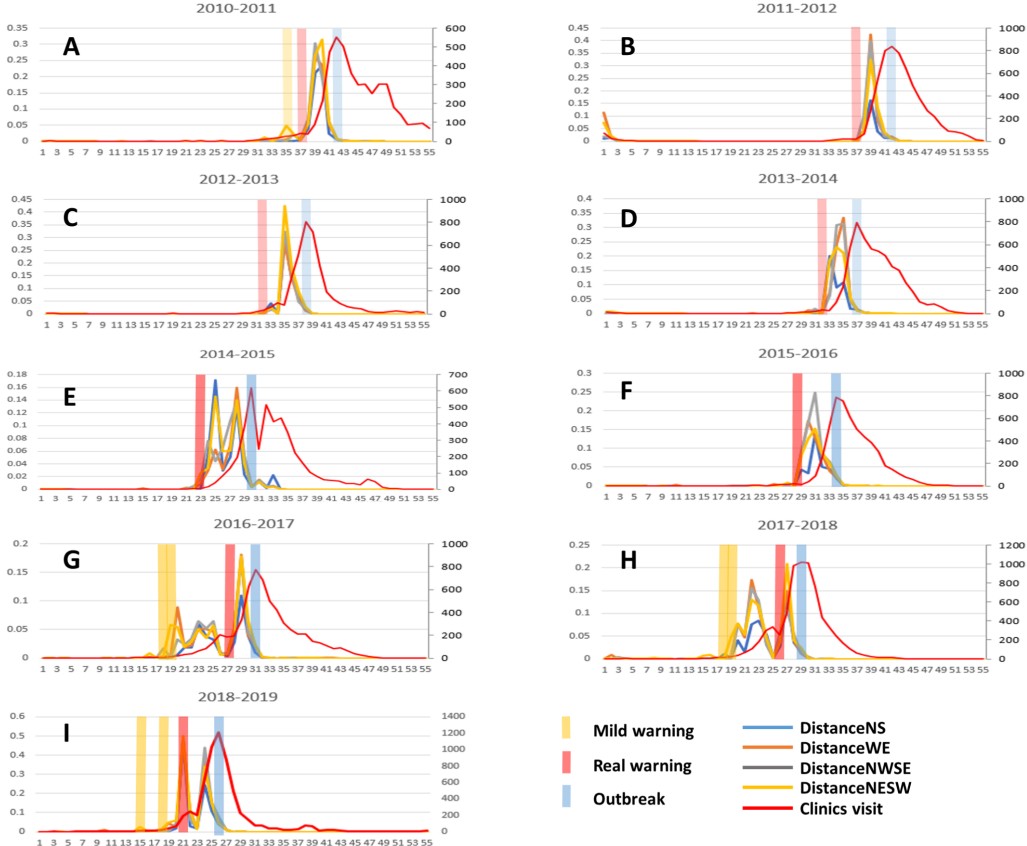

**Figure 4** **The predictions of annual influenza outbreak in Tokyo from 2010 to 2019.** Our SP-DNM method timely detect the early-warning signal of influenza outbreak for each year, i.e., results respectively in (A)2010-2011, (B)2011-2012, (C)2012-2013, (D)2013-2014, (E)2014-2015, (F)2015-2016, (G)2016-2017, (H)2017-2018, (I)2018-2019. The red line represents the clinic-visiting count, while the other lines represent the SP-DNM scores. $X$ axis represents the time evolution (week in a year) in a year. $Y$ axis on left side represents SP-DNM scores, and $Y$ axis on right side represents the clinic-visiting count. The yellow bar stands for mild warning, the red bar represents real warning and the blue bar represents the influenza outbreak point. To avoid violent disturbance when calculation window was too short, we use more data in last year to calculate Pearson's correlation coefficient and 55-week data including data in last year in calculation for unity.

To better show the effectiveness of our method, the diagram of the clinic-visiting counts was presented in Fig. 5, from which it is seen that our warning points (the red diamonds) were ahead of the outbreak points (the blue circles), revealing accuracy and effectiveness of SP-DNM method.

## Trace the change in the city network of Tokyo

To illustrate how SP-DNM works, we presented dynamic evolution of network in terms of weight from 2013 to 2014 at key period in Fig. 6. At the beginning of the year, it can be seen that all nodes are colored with light green and all edges are thin which means no influenza case and low correlations respectively. When system state approaches to its outbreak state, changes firstly occurred near node 2 at the 28th week, revealing that influenza are likely

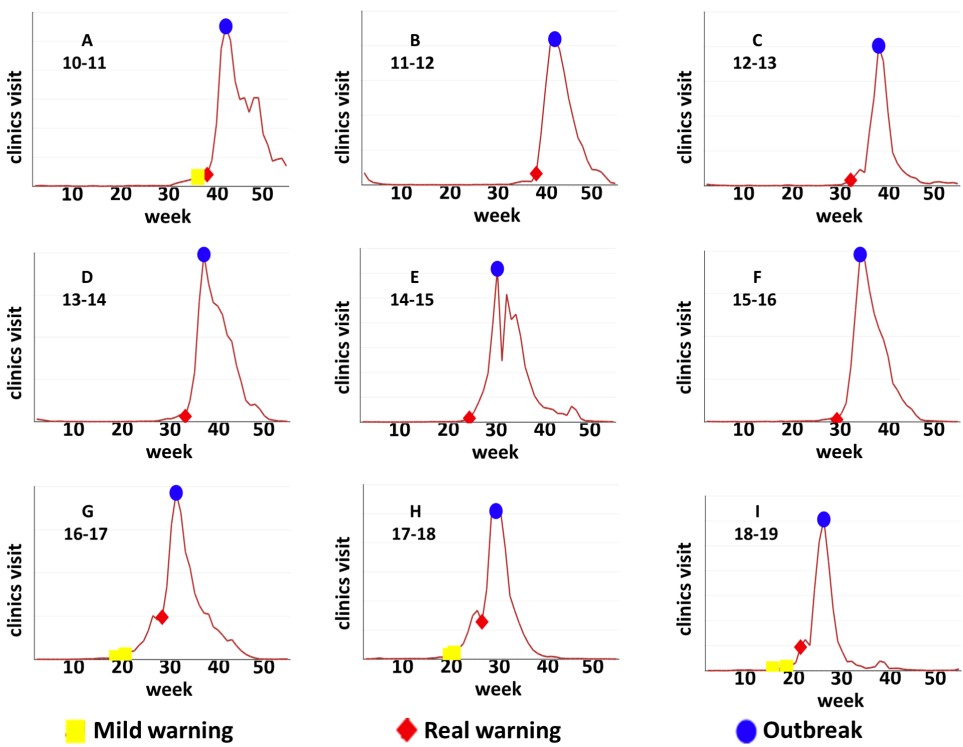

**Figure 5  Identified results by SP-DNM in Tokyo from 2010–2019.** It is clear that flu outbreak was accurately detected by the SP-DNM method for each year, i.e., results respectively in (A)2010-2011, (B)2011-2012, (C)2012-2013, (D)2013-2014, (E)2014-2015, (F)2015-2016, (G)2016-2017, (H)2017-2018, (I)2018-2019. The Y axis represents the number of clinic visits, the yellow rectangle represents the mild warning point, the red diamond represents real warning point, and the blue circle represents flu outbreak point.

to spread from region 2. As time goes by, the influenza impact gradually spread to farther region, leading more and more edges on the left side become thicker. Finally, both standard deviation and correlation between adjacent node increase sharply at the 32nd week before flu outbreak at the 36th week, causing an influence that the SP-DNM scores satisfied the 3-fold change threshold, which indicate the appearance of pre-outbreak state. To avoid those edge with high correlation, shortest path drastic change in the 29th and 30th week. However, at the 32nd week, most edges are in high correlation, so SP-DNM scores exceed three-fold threshold and the early-warning signal was generated. The dynamic change of network indicates that the SP-DNM method is able to represent real process state of influenza in the city network and generate early-warning signal of influenza, which is helpful for flu prevention. Moreover, dynamic evolution of other shortest paths in Tokyo city network were shown in the Supplemental Information.

## Detecting the pre-outbreak state in Hokkaido

To validate the generality and effectiveness in another region and another network, SP-DNM has also been applied in Hokkaido from 2009 to 2018. The detailed results were shown in the Supplemental Information. The early-warning signals were detected correctly and there was an average of 6-week window ahead without drastic fluctuation, which was
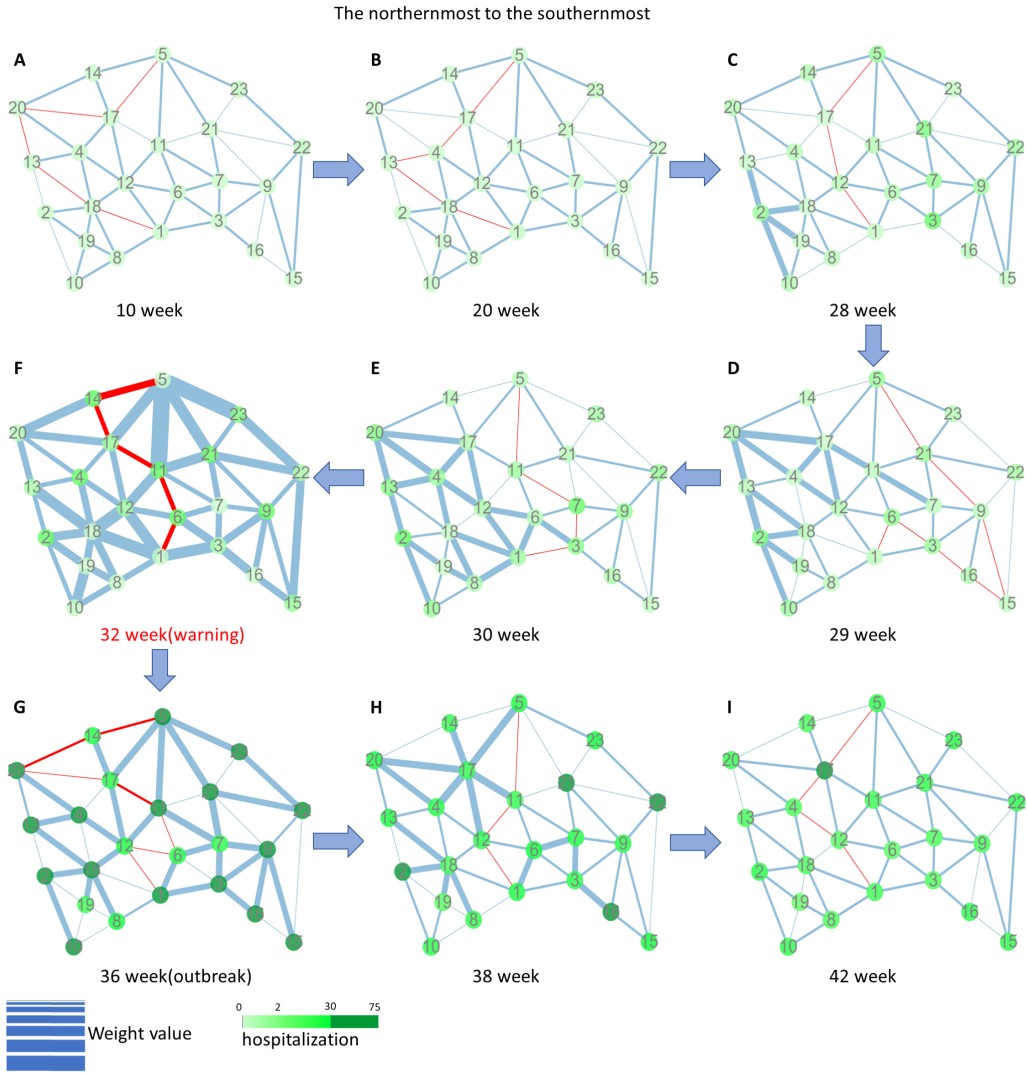

The northernmost to the southernmost

**Figure 6** **The dynamic evolution of Tokyo city network during a key period from 2013 to 2014.** It is clear that flu outbreak was accurately detected by the SP-DNM method, reflecting in (A) the 10th week, (B) the 20th week, (C) the 28th week, (D) the 29th week, (E) the 30th week, (F) the 32nd week, (G) the 36th week, (H) the 38th week, (I) the 42nd week. The *Y* axis represents the number of clinic visits, the yellow rectangle represents the mild warning point, the red diamond represents real warning point, and the blue circle represents flu outbreak point.

caused by the differences of the city networks between Tokyo and Hokkaido. There are 23 regions with 53 edges in Tokyo city network while 30 regions but only 49 edges in Hokkaido city network. And the transportation in Tokyo is much denser than that in Hokkaido. The population of Tokyo is much larger than that of Hokkaido. These factors caused that the flu transmission in Tokyo may be faster than in Hokkaido, and the performance of our method in Tokyo is better than that in Hokkaido.

## DISCUSSION

In the discussion, we make a comparison with our SP-DNM method, the sum of weight method (a method which just utilize weight of edges simply) and the Landscape DNM method (*Chen et al., 2019*). Moreover, an extra experiment has been performed to figure out the best number of shortest paths by deleting or adding the shortest paths.

### Evaluation of accuracy and stability

Before comparison, it is necessary to define metrics which measure accuracy and stability of predictive method.

- Accuracy rate. This metric gives the percentage of years which predict point is in front of outbreak point in all years. When a predict point was behind an outbreak point, it means prediction fail. It can be easily measured by

$$p = \frac{n_{right}}{n}$$

where $n_{right}$ represents the number of years which predict point is in front of outbreak point, $n$ represents the number of years.

- S-index (instability/error index). Based on the accuracy rate, our predicted point is supposed to possess a suitable range ahead outbreak point. S-index (instability /error index) can be measured by the mean-square error (MSE) by

$$S-\text{index} = \frac{\sum_{i=1}^{n}(w_i - w)^2}{n}$$

where $w_i$ represents the interval window that the predict point ahead of outbreak point in each year, $w$ represents the suitable window set according to historical data and region characteristic ahead outbreak point, $n$ represents the number of years. In our work, $w$ was defined by an average of difference between the week with fastest increase in clinic visits and outbreak week in each year. By the way, $w$ in Tokyo is 41/9 and $w$ in Hokkaido is 59/9 in our experiment. According to the definition of S-index, therefore, the lower the S-index is, the higher stability a method would be.

### Comparison with method using sum of weight

According to DNM theory, sum of weight can also regard as indicator of pre-outbreak state and we call the method utilizing sum of weight as indicator sum of weight method. However, this method is not suitable to detect pre-outbreak states for some reason, due to the sensitivity of Pearson correlation coefficient when the data is perturbed by noises, which would be likely to amplify weight of some edges, causing appearance of fake signal. Besides, it neglects spatial feature of network, leading to untimely signal, which leads to the difficulty of studying diseases transmission and nodes interaction. Results of comparison were shown in Table 1. Clearly, the warnings of sum of weight method are earlier than that of our method, causing more drastic fluctuation and higher S-index. In Fig. 6, from 2013 to 2014, at the 29th week, edges on the left side become thick, in this case, the surveilling system based on the sum of weight method generated signal, which resulted in a bigger fluctuation before the outbreak. While at the 32nd week, nearly all edges become thick,

**Table 1  Comparisons with other methods on detecting flu outbreak in Tokyo in all years.** It includes 2-shortest-path DNM method, 4-shortest-path DNM method, 6-shortest-path DNM method, sum of weight DNM method and landscape DNM method. The number in the bracket indicates the difference between the prediction point and the outbreak point.

| Time | 10-11 | 11-12 | 12-13 | 13-14 | 14-15 | 15-16 | 16-17 | 17-18 | 18-19 | Accuracy | S-index |
|---|---|---|---|---|---|---|---|---|---|---|---|
| 2 shortest paths (week) | 35(6) | 37(4) | 31(6) | 32(4) | 22(7) | 28(5) | 19(11) | 19(9) | 19(7) | 100% | 8.691 |
| 4 shortest paths (week) | 37(4) | 37(4) | 31(6) | 32(4) | 23(6) | 28(5) | 27(3) | 25(3) | 21(5) | 100% | 1.148 |
| 6 shortest paths (week) | 37(4) | 37(4) | 31(6) | 32(4) | 23(6) | 28(5) | 27(3) | 25(3) | 21(5) | 100% | 1.148 |
| Sum of weight (week) | 37(4) | 32(9) | 30(7) | 29(7) | 22(7) | 23(10) | 17(13) | 18(10) | 15(11) | 100% | 23.35 |
| Landscape DNB (week) | 30(11) | 34(7) | 32(5) | 32(4) | 25(4) | 26(7) | 20(10) | 18(10) | 17(9) | 100% | 14.814 |
| Outbreak (week) | 41 | 41 | 37 | 36 | 29 | 33 | 30 | 28 | 26 | ____ | ____ |

which confirms the actuality of the generated signal. Because those weights on the left side increase sharply at the 29th week, exploiting sum of weight method should consider such an influence (generate signal 7-weeks ahead) while SP-DNM method is not likely to be influenced by this significant change (generate signal 4-weeks ahead). In conclusion, the SP-DNM method is relatively more robust than the sum of weight method, and additionally exploiting our method can effectively avoid generating fake signal.

## Comparison with Landscape DNM method

A local landscape DNM method was proposed to detect the critical state of flu outbreak (*Chen et al., 2019*), which combined $PCC_{in}$, $PCC_{out}$ and $SD_{in}$ as an indicator to describe the flu spread and outbreak. We applied landscape DNM method on Tokyo, the comparison result was shown in Table 1. It is seen that signals are about 4 to 11 weeks ahead the flu outbreak when using landscape DNM method to detect the pre-outbreak state, which is with large fluctuation. Moreover, S-index of SP-DNM are lower than Landscape DNM method's, which shows the high stability of our method.

## Comparison with logistic regression

In our study, we proposed a computational method, the shortest-path-based dynamical network marker (SP-DNM), which leverages the graph theory to detect the early warning signal of influenza outbreak. A comparison was carried out between the proposed SP-DNM and traditional logistic regression as shown in Fig. 7. It is seen that the performance of SP-DNM is better than logistic regression. The AUC of SP-DNM-based surveillance system is 0.8978 and the AUC of logistic-regression-based system is 0.8396. It should be noted that the monitoring system solely use counts of clinic visits. In addition, our method is a model-free method, which means that there is no training and testing modules in analysis, and thus avoiding the overfitting problem.

## The best number of shortest paths

An extra analysis was carried out to figure out the best number of shortest paths and result was listed in Table 1. Two original paths (node 20 to node 15, node 23 to node 2) were deleted, forming a two shortest path experiment. Two additional paths (node 2 to node 16, node 14 to node 18) were added, forming a six-shortest-path experiment. Compared with four-path SP-DNM method, signals were detected 1 or 2 weeks ahead in 2010–2011,

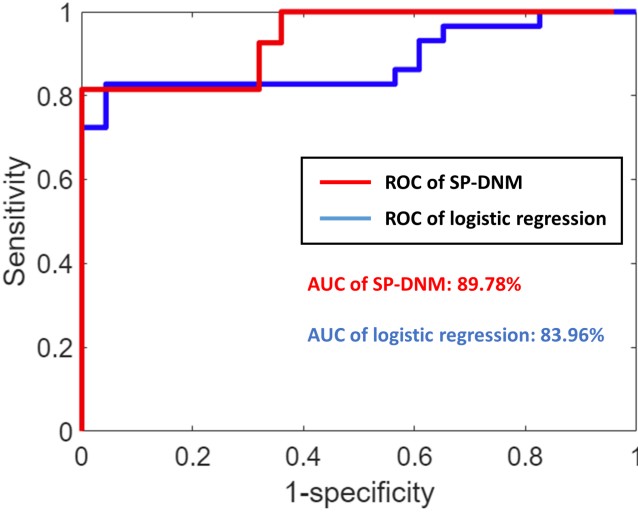

**Figure 7  The performance comparisons of SP-DNM-based and logistic-regression-based methods.**
The AUC of SP-DNM is 0.8978, while that of logistic regression is 0.8396. It can be seen that the SP-DNM-based surveillance system performs better than logistic regression.

2014–2015 and 2018–2019, 6 and 8 weeks ahead in 2016–2017, 2017–2018 with two-path SP-DNM method. And result of six-path SP-DNM method is the same as four-path SP-DNM method. When the number of shortest paths decreases, our method cannot cover the whole network causing that 3-fold change threshold conditions are tailless to satisfy, which make an earlier signal. When the number of paths increases, the calculation result keeps invariant, because four shortest paths have already covered the whole network. For Tokyo or Hokkaido, four-path SP-DNM method is enough for detecting pre-outbreak states. Nevertheless, when quantity of nodes increases or network structure changes, SP-DNM may require more shortest paths.

## CONCLUSIONS

Influenza is a complex disease threatening people's lives and causing huge economic losses. Due to its complicated biological system and the equally intricate external environment, it's challenging to predict influenza outbreak according to similar phenotype and outer data expression. It is of great importance to find an efficient and effective method to detect the outbreak of influenza, so as to make timely and appropriate intervention or prevention for the catastrophic epidemics, saving economic losses and human lives.

In our work, a new computational method SP-DMN based on the dynamical change of shortest paths in a city network was proposed to detect the pre-outbreak state of influenza in dynamic city network, which performed well in Tokyo and Hokkaido. By using this method, pre-outbreak states can be predicted with an average of 4-week window ahead, providing enough time for making proactive strategy to control influenza.

Comparing with the traditional DNM approach, there are some advantages of our proposed method. First, the SP-DNM is a model-free method, which means that there is no

training and testing modules in analysis, and thus avoiding the overfitting problem. Because our method solely depends on the statistical indicators including Pearson correlation coefficient and standard deviation. Second, SP-DNM has strong resistance to noise and can avoid fake signal, thus detecting pre-outbreak states accurately. Third, we can study disease transmission and nodes interaction by watching dynamic change of network. Our classification system of influenza spread helps to make exact and effective anti-influenza strategy.

However, there are still limitations in SP-DNM. When the connectivity of the graph becomes extremely small, our method performed approximate the effect of the sum of weight method due to the shortest path's rare changes as the evolution of weighted city network. Also, for different structures of the network, a different quantity of shortest paths may influence the prediction result, which is our future research target.

## ACKNOWLEDGEMENTS

The authors are grateful to Professor Yongjun Li for the discussion.

### Funding

This work was supported by National Natural Science Foundation of China (Nos. 11771152, 11901203, 11971176), Guangdong Basic and Applied Basic Research Foundation (2019B151502062), China Postdoctoral Science Foundation funded project (No. 2019M662895) and the Fundamental Research Funds for the Central Universities (2019MS111). The funders had no role in study design, data collection and analysis, decision to publish, or preparation of the manuscript.

### Grant Disclosures

The following grant information was disclosed by the authors:
National Natural Science Foundation of China: 11771152, 11901203, 11971176.
Guangdong Basic and Applied Basic Research Foundation: 2019B151502062.
China Postdoctoral Science Foundation funded project: 2019M662895.
Fundamental Research Funds for the Central Universities: 2019MS111.

### Competing Interests

The authors declare there are no competing interests.

### Author Contributions

- Yingqi Chen conceived and designed the experiments, performed the experiments, prepared figures and/or tables, authored or reviewed drafts of the paper, and approved the final draft.
- Kun Yang and Jialiu Xie performed the experiments, authored or reviewed drafts of the paper, and approved the final draft.
- Rong Xie and Zhengrong Liu analyzed the data, authored or reviewed drafts of the paper, and approved the final draft.

- Rui Liu conceived and designed the experiments, prepared figures and/or tables, authored or reviewed drafts of the paper, and approved the final draft.
- Pei Chen conceived and designed the experiments, prepared figures and/or tables, and approved the final draft.

## Data Availability

The data and code are available in a Supplemental File.

## Supplemental Information

Supplemental information for this article can be found online at http://dx.doi.org/10.7717/peerj.9432#supplemental-information.

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
