# Peer review of "Detecting the outbreak of influenza based on the shortest path of dynamic city network"

_PeerJ, doi:10.7717/peerj.9432_

## Round 0.1 · original submission · Minor Revisions

The reviewers appreciate the timing and important study for a new computational framework to forecast the outbreak of flu. They also provide prompt and helpful feedback. Please address their concern and incorporate their suggestions and submit a revision ASAP.

Reviewer 1 ·

Basic reporting

In this paper, the authors proposed a method for identifying the dynamic network marker of influ outbreak. This is an important and challenging research topic. The number of the patients are not fully collected and some of the most important literature have not been cited. The prediction results are not clearly presented.

Experimental design

The experiments of building up a concept of dynamic network marker for influenza outbreak need be modified to de novo predictions. It is a time series and sequential data prediction. The sliding prediction with the time points need be implement to show the efficiency of the proposed method.

Validity of the findings

As claimed in the former comments, the validity of the results need be justified by more experiments. Moreover, the authors have cited some of the existing method. It is not clear what is the difference and improvement of this method compared to the other existing ones. For instance, the methods described in Ref. 14. Moreover, it is not clear the method listed in the other similar references Ref. 20-24.

Additional comments

In this paper, the authors developed a method for predicting the outbreak of influenza. This is an important topic for public health. Beyond the other comments listed above, I listed some comments here for consideration. First is about the network. The paper claimed the network model. However, it is not clear about the network property, such as the node and edge. The graph should be complete. The present graph is strange. What is the prediction? If only using the number of recorded patients, the traditional methods such as autoregression and many others in statistics can do this job. And the comparison with them is expected to show the necessity of building up a new method. Second is about the dynamics. In my understanding, the numbers of every hospital/city is dynamic and the movements of them and the linkage between the cities are also time dependent. It is not very clear how to include this important dynamics in this prediction. Third, the presentation need be improved. For instance, the discussion seems to be another result. The language need further polished before publication.

·

Basic reporting

In figure 3b, the columns may be mislabeled. Please carefully examine the column names.

Experimental design

no comments

Validity of the findings

no comments

Additional comments

The authors propose a new computational framework to forecast the outbreak of flu. It is interesting for the finding that changes of shorted paths between two points of a weighted city network are predictive of a flu outbreak. Two minor concerns should be further clarified before publication.

1. The significance of the finding should be further discussed. In particular, the authors should sufficiently discuss the advantage of the new finding over traditionally used metrics such as the incidence of flu.

2. "outbreak" can be further quantified, i.e., the quantitative relationship between the signals and the "outbreak" consequences can be further quantitatively evaluated.

·

Basic reporting

In this study, the geographical districts/cities, Hokkaido and Tokyo, are in Japan. Please mention in the text that Hokkaido and Tokyo are located in Japan.

Line 38: Please revise “in the U.S.”

Line 166: Please revise “was based on”.

Lines 186-187: It says: ”Both the two datasets have been normalized by dividing the number of wards in a region.” In Figure 1 (A), it says “Normalized by: data (t) = total hospitalization count / clinics count”. Please check and clarify the terms: “clinic count”, “clinic visits”, “hospitalization”, and “wards”. When speaking of health care systems with possible differences between various countries worldwide, are these “clinics” hospitals (specialized care: wards? outpatient clinics?) and/or primary care encounters (health care centers?) or what kind of clinics? Please consider describing briefly the Japanese health care system in terms of these “clinics”.

Lines 198-202: There are different paths to nodes (Kita-ku, Minato-ku, Suginami-ku, Katsushika-ku, Nerima-ku, Edogawa-ku, Adachi-ku, Setagaya-ku). Please consider using bold font or italics with these Japanese words.

Line 220: Please revise “seen in Figure 4 that”.

Lines 281 and 219: Please revise “based on”.

Line 339: Please revise “human lives”.

Both “clinic visits” and “clinics visits” are used in the text. Please choose “clinic visits”.

Figure 1: This Figure 1 seems to be in the end of the manuscript but Figure 1 is not mentioned in the text at all.

Figure 1, legend: It says: “covering 53 districts…”. However, in Figure 1 (A), there are 23 districts and 30 cities mentioned. Are these districts or cities? Please revise.

Figure 1 (A): A typo “peopele”. Please revise “people”.

Figure 1 (C): The text around the pattern is too small to be read.

Figure 3: The first line of the table B: The title of regions 13-23 seems to be wrong. Please change the order between “Region” and “Code”.

Figure 4: X axis seems to be in weeks. Please mention this in the figure legend.

Figure 4: X axis seems to end in week 55. Typically, the year ends in weeks 51-53. Please revise all X axises during 2010-2019 by checking that X axis ends in the exact week when the year ends.

Figure 4, legend: Please revise “on left/right side”.

Figure 5: There are two terms “hospitalization” and “clinic visits”, included elsewhere in the text as well. Are these the same? Health care systems in different countries may vary, thus the terms need to be clarified. If the patient comes to a health care unit and is sent back home, is this a “clinic visit”?, and if the patient is taken into the hospital ward, is this “hospitalization”? Or are “clinic visit” and “hospitalization” the same thing?

In Figure 5, legend, and also elsewhere in the text: How do you determine “the number of clinic visits”? Are they the patients with influenza (or influenza-like illness) symptoms, and then a physician reports influenza diagnoses and/or laboratory positive findings of influenza A or B? Please clarify.

In Figure 5, please choose “blue bullet” instead of “blue circle”.

Experimental design

The research question is relevant and meaningful, although you may consider using the following to describe the purpose of your study: “The aim of this study was…” in the end of the Introduction and also in the Abstract.

Validity of the findings

This study has public health impact.

Additional comments

Thank you for having me to review this manuscript. This is a study on detecting the outbreak of influenza based on the shortest path of dynamic city network. My major revision requests include the terms related to the health care system.

Reviewer 4 ·

Basic reporting

The language of the paper is English and clear, unambiguous and technically correct text. The cited references in the paper are correct and sufficient enough to support the research background/context. The structure of the paper is possible to obey to the format, but I suggest that the figures and tables should be placed to be right for reading the paper. The The relevant results are self-contained in the paper. And the discussion is given between the results and conclusions. Therefore, I think that the paper is satisfied with the requirement of your journal.

Experimental design

The paper is within aims and scope of the journal.
The research question is clear, relevant and meaningful. The proposed method is detect the pre-outbreak state of influenza epidemics by monitoring the dynamical change of the shortest path in a city network to fill the gap. This method gives the sufficient detail and the experiments have done sufficiently.

Validity of the findings

The findings of the paper is valid. According to the figures and the tables, the underlying data have been provides. The conclusions are given and are relevant to the results.

Additional comments

This paper proposed a computational method, the shortest-path-based dynamical network marker (SP-DNM), to detect the pre-outbreak state of influenza epidemics by monitoring the dynamical change of the shortest path in a city network. The idea, structure and quality of the paper are acceptable and the authors did a high quality research on detecting the outbreak of influenza. There are some points before acceptance:
1. In Figure 2, the terminal condition of algorithm is missing.
2. The figures and tables should be the right place.
Before this paper is published, I would suggest that the wording, sentence structure and
grammatical structure of the article should be improved.
I recommend that this paper is published in your journal.

---

## Round 0.2 · accepted · Accept

All the reviewers appreciate your efforts to an improved version and are satisfied with the revision. Now I suggest its acceptance after adding some discussion on how to use the identified markers in a real condition.

Reviewer 1 ·

Basic reporting

no comment.

Experimental design

no comment.

Validity of the findings

No comment.

Additional comments

I appreciate the responses to my former comments. I am concerning about how to use the identified markers in a real condition. For instance, in the city of Tokyo as reported. If you find some markers from the data, how to use it and end this transmission/outbreak of the disease. The authors need consider useful and specific application for these important findings that is important for this kind of research, especially in the pandemic of COVID-19.

·

Basic reporting

The manuscript is well written, with the whole story and related literature references clearly stated. The results are also self-contained.

Experimental design

The research question is well defined. Basically, this research adds a new dimension to forecast the outbreak of influenza in the increasing crowd city space.

Validity of the findings

The finding is interesting, which may also be applied to other infectious diseases given the related data. Particularly, with rapid global development, the results are relevant to preventing and forecasting new outbreaks of infectious diseases.

Additional comments

The authors have addressed all my previous concerns.

·

Basic reporting

.

Experimental design

.

Validity of the findings

.

Additional comments

.

Reviewer 4 ·

Basic reporting

The article is clear and unambiguous,professional English used thoughout. The adopted references are enough. And the professional article structure,figures,tables,raw data are shared.

Experimental design

The content of the paper is within Aims and Scope of the journal.

Validity of the findings

The results of the paper are shown that the proposed method is novel and the conclusions are well stated.

Additional comments

This paper proposed a computational method, the shortestpath-based dynamical network marker (SP-DNM), to detect the pre-outbreak state of influenza epidemics by monitoring the dynamical change of the shortest path in a city network. The questions of last review are solved.
I agree that this paper will be published in Peer J.